# Does the TT Variant of the rs966423 Polymorphism in *DIRC3* Affect the Stage and Clinical Course of Papillary Thyroid Cancer?

**DOI:** 10.3390/cancers12020423

**Published:** 2020-02-12

**Authors:** Kinga Hińcza, Artur Kowalik, Iwona Pałyga, Agnieszka Walczyk, Danuta Gąsior-Perczak, Estera Mikina, Tomasz Trybek, Monika Szymonek, Klaudia Gadawska-Juszczyk, Klaudia Zajkowska, Agnieszka Suligowska, Artur Kuchareczko, Karol Krawczyk, Janusz Kopczyński, Magdalena Chrapek, Stanisław Góźdź, Aldona Kowalska

**Affiliations:** 1Molecular Diagnostics, Holycross Cancer Centre, S. Artwińskiego Str. 3, 25-734 Kielce, Poland; arturko@onkol.kielce.pl (A.K.); karol.grzegorz.krawczyk@wp.pl (K.K.); 2Collegium Medicum, Jan Kochanowski University, IX Wieków Kielc Av. 19, 25-319 Kielce, Poland; iwonapa@tlen.pl (I.P.); stanislawgo@onkol.kielce.pl (S.G.); aldonako@onkol.kielce.pl (A.K.); 3Endocrinology, Holycross Cancer Centre, S. Artwińskiego St. 3, 25-734 Kielce, Poland; a.walczyk@post.pl (A.W.); danutagp@o2.pl (D.G.-P.); esterami@tlen.pl (E.M.); trytom1@tlen.pl (T.T.); christ76@interia.pl (M.S.); klaudiagdwsk@op.pl (K.G.-J.); klaudia.ziemianska@gmail.com (K.Z.); a.suligowska@wp.pl (A.S.); arturkuchareczko@gmail.com (A.K.); 4Surgical Pathology, Holycross Cancer Centre, S. Artwińskiego Str. 3, 25-734 Kielce, Poland; januszko@onkol.kielce.pl; 5Department of Probability Theory and Statistics Institute of Mathematics, Faculty of Mathematics and Natural Sciences, Jan Kochanowski University, Świętokrzyska Str. 15, 25-406 Kielce, Poland; Magdalena.Chrapek@ujk.edu.pl; 6Clinical Oncology, Holycross Cancer Centre, S. Artwińskiego Str. 3, 25-734 Kielce, Poland

**Keywords:** papillary thyroid cancer, overdiagnosis, overtreatment, risk factors

## Abstract

Thyroid cancer (TC) is the most common cancer of the endocrine system. Most new diagnoses are of low-grade papillary thyroid cancer (PTC), suggesting that PTC may be over-diagnosed. However, the incidence of advanced-stage PTC has increased in recent years. It is therefore very important to identify prognostic factors for advanced PTC. Somatic mutation of the *BRAF* gene at V600E, or the coexistence of the *BRAF* V600E mutation and mutations in the *TERT* promoter are associated with more aggressive disease. It would also be valuable to identify genetic risk factors affecting PTC prognosis. We therefore evaluated the impact of the rs966423 polymorphism in the *DIRC3* gene, including its relationship with unfavorable histopathological and clinical features and mortality, in differentiated thyroid cancer (DTC). The study included 1466 patients diagnosed with DTC from one center. There was no significant association between the *DIRC3* genotype at rs966423 (CC, CT, or TT) and any histopathological or clinic factor examined, including initial response to therapy, response at follow-up, or overall mortality, in DTC patients.

## 1. Introduction

Thyroid cancer (TC) is the most common cancer of the endocrine system, accounting for 1–2% of all malignancies. The World Health Organization reported 567,233 new cases of TC in 2018, of which 130,889 occurred in men and 436,344 occurred in women [1]. Recent years have seen a very sharp rise in incidence of differentiated thyroid cancer (DTC), mainly low-grade papillary thyroid cancer (PTC). The increase in the diagnosis of PTC is thought to result from better availability of diagnostic imaging and the frequent use of fine needle aspiration biopsies. Moreover, the proportion of microcarcinomas found during cancer surgery increased from 18.4% in 1983–1987 to 43.1% in 1998–2001 [2]. Autopsy studies show that papillary thyroid microcarcinoma was found in up to 35.6% of people who died from other causes [3,4]. PTC is a relatively benign tumor, in which the 10-year survival rate is approximately 95% [5]. Therefore, it is possible that PTC is over-diagnosed and over-treated [5,6]. However, a study of the Surveillance, Epidemiology, and End Results (SEER) dataset examining 77,276 patients with TC from 1974–2013 identified an increase in the incidence of advanced-stage PTC [7]. The main risk factors that have led to this worldwide increase in TC are obesity, smoking, iodine intake, and environmental risk factors such as exposure to radiation, flame retardants, or volcanic ash [8]. The overall mortality of TC varies over time. According to a study of the SEER-9 data, the overall incidence of thyroid cancer rose by 3.6% per year between 1974 and 2013, and incidence-based mortality increased by 1.1%. Overall mortality increased from 0.40 per 100,000 person-years in 1994–1997 to 0.46 per 100,000 person-years in 2010–2013 [7,8]. At the same time, there are patients in whom the course of the cancer is rapid, and early identification of these patients is extremely important for treatment efficacy. So far, a more severe disease course has been associated with unfavorable histopathologic features, the presence of the V600E somatic mutation of the *BRAF* gene, and the coexistence of the *BRAF* V600E mutation with mutations in the *TERT* promoter [9,10]. However, all these factors can be identified only after surgical intervention, which makes it challenging to properly plan and adapt the treatment to the needs of a particular patient. It would be very attractive if germline genetic risk factors could be identified, which would allow physicians to estimate the prognosis of the patient before surgery. In contrast to somatic changes, which can only be identified in surgically obtained tumor samples, germline variants can be identified in the peripheral blood of the patient. Research carried out to date identified a few genes associated with vulnerability to the development of DTC, including *DICER1, FOXE1, PTCSC2, MYH9, SRGAP1, HABP2, BRCA1, CHEK2, ATM, RASAL1, SRRM2, XRCC1, TITF-1/NKX2.1,* and *PTCSC3* [11,12,13,14,15,16,17,18,19,20,21,22]. Recently, seven chromosomal loci (1p13.2–1q21, 1q21, 2q21, 6q22, 8p23.1–p22, 8q24, and 19p13.2) associated with DTC development were identified [18]. Genome-wide association studies (GWAS) identified genetic variants as candidate germline risk factors in Japanese, Chinese, Polish, and British populations [23]. In the literature, there are a few reports demonstrating the prognostic significance of the rs966423 polymorphism in the *DIRC3* (disrupted in renal carcinoma 3) gene, and its relationship with unfavorable histopathologic features and mortality in patients diagnosed with DTC. The rs966423 polymorphism is located at the 2q35 locus of *DIRC3*, within a long non-coding RNA (lncRNA). *DIRC3* was first identified in 2003 as a fusion transcript involved in familial renal carcinoma [24]. Due to the ambiguous role of the rs966423 polymorphism in *DIRC3* as a prognostic factor in PTC, we assessed the correlation between this variant and the clinical course of TC. 

## 2. Materials and Methods

### 2.1. Patients and Controls

All study procedures were approved by the Institutional Review Board of the Holycross Chamber of Physicians in Kielce, Poland. Living patients provided signed consent for molecular tests to be carried out. The study included 1466 patients (1445 alive patients and 21 dead patients, where types of TC were following: 1386 PTC cases, 42 poorly differentiated thyroid carcinoma (PDTC) cases, 34 follicular thyroid carcinoma (FTC) cases, and 4 oxyphilic TC cases) diagnosed between 2000 and 2018 at the Holycross Cancer Center (HCC) in Kielce and patients who were diagnosed with PTC in 2000–2018 but whose monitoring was stopped due to death. A group of 309 healthy individuals was included in the study as a control cohort. All cases and controls were recruited at the HCC in Kielce. Peripheral blood was taken, and archived blocks of primary tumor tissue were obtained. Patients who failed to provide signed consent for molecular testing or who were lost to follow-up for reasons other than patient death were excluded. Retrospective clinical data were available for all of the cases. Demographic and clinical data, including patient gender, age at diagnosis of TC, tumor size, multifocality, extrathyroidal extension, vascular invasion, lymph node, distant metastasis, clinical stage, response to primary treatment, clinical course, and final disease status were obtained from medical records. All cases were restaged according to the recently updated 8th Edition of the American Joint Committee on Cancer (AJCC) tumor node metastasis (TNM) Staging System. 

### 2.2. DNA Isolation from Blood Samples

Genomic DNA isolated from whole peripheral blood was used for genetic analyses. Material was isolated using the Maxwell RSC Blood DNA Kit (Promega, Madison, WI, USA), according to the manufacturer’s instructions. DNA quality and purity were verified using a NanoDrop spectrophotometer (Thermo Fisher Scientific, Waltham, MA, USA). 

### 2.3. DNA Isolation from FFPE Samples

A pathologist marked the area without tumor cells on hematoxylin and eosin-stained slides. The pathologist-selected areas from unstained slides were used for DNA extraction using the Maxwell 16 FFPE Tissue LEV DNA Purification Kit, according to the manufacturer’s instructions (Promega). DNA quality and purity were verified on a NanoDrop spectrophotometer (Thermo Fisher Scientific).

### 2.4. Quantitative Polymerase Chain Reaction (qPCR) and Sanger Sequencing

Detection of *DIRC3* genetic variants was performed by quantitative polymerase chain reaction (qPCR) using the TaqMan SNP Genotyping Assay kit (Thermo Fisher Scientific) and sequencing of PCR products. The region of *DIRC3* containing the polymorphism was amplified using the following oligonucleotide primer pair: forward, 5′-CAGCCTTTCATCCAGCAGGACAACAG-3′ and reverse, 5′-TCCACTGGGCGTCTCAACTACAATCTG-3′. The amplification conditions consisted of an initial denaturation at 95 °C for 15 min and then 39 cycles of denaturation at 95 °C for 15 s, annealing at 61 °C for 15 s, and extension at 72 °C for 15 s, followed by a final extension at 72 °C for 2 min. PCR was performed using a Veriti Thermal Cycler (Applied Biosystems, Foster City, CA, USA). PCR products were separated by microchip electrophoresis using the MultiNA System (Shimadzu Corporation, SHIM-POL, Poland). Amplified PCR products were purified using FastAP and Exonuclease I (Thermo Fisher Scientific). After purification of the resulting PCR products, sequencing was performed using the BigDye Terminator v1.1 Cycle Sequencing kit (Life Technologies/Thermo Fisher Scientific), and the same forward and reverse primers. The primers were diluted at a ratio of 8:42 μL in water, and PCR products were separated and analyzed on an ABI 3130 Automatic Capillary DNA Sequencer (Applied Biosystems).

### 2.5. Statistical Analysis

Quantitative data are described by means, standard deviations, medians, quartiles, and ranges (minimum and maximum). Categorical data were described by frequencies and percentages. Group comparisons were performed using chi-squared or Fisher’s exact tests for categorical variables; t-tests for quantitative, normally distributed variables; or Mann–Whitney tests for quantitative, non-normally distributed variables. The normality of distribution was determined by Shapiro–Wilk tests. Statistical tests were two-tailed, and *p*-values < 0.05 were considered significant. All statistical analyses were performed using the R software package, version 3.6.1., The R Foundation for Statistical Computing, Vienna, Austria.

## 3. Results

### 3.1. Characteristics at Presentation and Primary Treatment

Patient demographics and the clinicopathological features of all 1466 cases are presented in (Table 1, Appendix A). There were 1269 women (86.6%, 1269/1466) and 197 men (13.4%, 197/1466) in the study group. The study group included patients with the four most common types of TC: 1386 PTC cases (94.5%), 34 FTC cases (2.3%), 42 PDTC cases (2.9%), and four oxyphilic cases (0.3%). The dominant histological variant was classic for PTC (71.2%), minimally invasive for FTC (85.3%), and insular for PDTC (69%). The median age at diagnosis was 51 years (range, 15–85), and most patients (61.8%) were < 55 years old. The median tumor size was 9 mm (range, 0.0–130). Most patients (74.2%, 1088/1466) did not have tumor multifocality. Gross type extrathyroidal extension was detected in 1.6% patients (23/1466). Angioinvasion was present in 6.8% (99/1466) of patients. Lymph node metastases and distant metastases were present in 14.8% (218/1466) and 2.3% (34/1466) of patients, respectively. An advanced TNM stage of III or IV, according to the updated 8th Edition of the AJCC TNM staging system, was observed in 26 patients (1.8%, 26/1466). According to the American Thyroid Association (ATA) initial risk criteria, 959 (65.4%) patients were classified as low risk (LR), 416 (28.4%) as intermediate risk (IR), and 91 (6.2%) as high risk (HR).

### 3.2. Response to Therapy and Follow-up

The initial response to therapy was evaluated for 12 months at our center. The types of responses were classified into 4 groups: excellent, indeterminate, biochemical incomplete, and structural incomplete according to the criteria in the ATA recommendations [25]. Out of 1466 patients, 1343 were assessed. For 123 patients, no initial response to therapy was available because the course of treatment was too short. An excellent response to primary therapy was observed in 1116 patients (83.1%), while in 160 patients (11.9%) the response was indeterminate. A biochemical incomplete response to initial therapy was observed in 32 patients (2.4%), while a structural incomplete response was observed in 35 patients (2.6%). Recurrence after an excellent response to initial therapy occurred in ten patients (0.7%). Recurrence was defined as the appearance of disease after no evidence of disease (NED). At the end of the study, NED was observed in 1226 patients (91.3%). Of the remaining patients, 69 (5.2%) had an indeterminate response and 47 patients (3.5%) exhibited biochemically or structurally persistent disease. There were three deaths among cases with the classic variant of PTC, four deaths among cases with the solid variant of PDTC, 13 deaths among cases with the insular variant of PDTC, and one death among cases with the oxyphilic type of TC. The median follow-up time of the study group was 60 months (range: 24–132 months) (Table 2, Appendix A).

### 3.3. Relationship between rs966423 Genotype and Histopathologic Factors and Response

The frequencies of the rs966423 genotypes (CC, CT, and TT) in the study group were 22% (322/1466), 49.8% (730/1466), and 28.2% (414/1466), respectively. In the control group, 17.5% (54/309) of patients were CC carriers, 50.2% (155/309) were CT carriers, and 32.4% (100/309) were TT carriers (Table 3). Statistical analysis revealed no significant difference in the frequencies of DIRC3 variants between the study group and the control group (Fisher’s exact test, *p* = 0.140). In addition, statistical analysis indicated no significant difference in the ratio of TT carriers versus CC or CT carriers in patients with TC compared with the control group (Fisher’s exact test, *p* = 0.147). We also assessed the relationships between polymorphisms in *DIRC3* (CC, CT, or TT) and clinical factors, including initial response to therapy, response at follow-up, and overall mortality. No statistically significant differences were found (Table 4). In addition, statistical analysis in ATA initial risk sub-groups indicated no significant difference in association the frequencies of the TT carriers and high ATA initial risk (Appendix A). However the highest risk of recurrence is observed in the first 5 years of follow-up [26], we did not find a significant association of the frequencies of recurrence or death in the TT carriers in analysis of 538 patients with longer follow-up (the median follow-up time: 13 years) (Appendix A).

## 4. Discussion

Genetic prognostic factors in DTC are still poorly understood, but increasing evidence suggests that multiple low-penetrance genetic variants, rather than a few high-penetrance variants, can better explain the risk of DTC [27]. A GWAS of European populations identified five low-penetrance variants potentially involved in DTC risk, including rs965513 on 9q22, rs944289 on 14q13, rs966423 on 2q35, rs2439302 on 8p12, and rs116909374 on 14q13.3 [28,29]. The rs966423 polymorphism is located at 2q35 of the *DIRC3* gene, within a lncRNA. *DIRC3* was first identified in 2003 as a fusion transcript involved in familial renal carcinoma [24]. Although the function of *DIRC3* is unknown, it is thought to have tumor suppressor activity. In a GWAS, *DIRC3* variants were associated both with TC risk and thyroid stimulating hormone levels. It is thus possible that genetic variants in *DIRC3* alter thyroid stimulating hormone production, and therefore indirectly promote TC development as a result of decreased thyroid epithelium differentiation [28]. Recently, Świerniak et al. published the results of a study indicating a relationship between the TT variant of the rs966423 polymorphism, and a worse prognosis and increased overall mortality in patients with DTC [30]. Świerniak et al. detected the TT genotype in 518/1836 patients (28%) in their cohort, but the study did not include a control group. We detected the TT genotype in 414/1466 DTC patients (28.2%) and 100/303 healthy individuals (32.4%) in the control cohort. Statistical analysis indicated no significant difference in the incidence of the TT variant between the DTC patients and the control population (Fisher’s exact test, *p* = 0.140). Similar data was published by Wang et al., where the TT genotype was detected in 2.7% of PTC patients and 4.3% of controls, although this study was conducted in a Chinese population. [31] These observations are inconsistent with the idea that the TT variant of rs966423 is a risk factor for the development of DTC. In addition, we assessed the correlation between *DIRC3* genotype (CC, CT, or TT) and histopathological and clinical features, including initial response to therapy, response at follow-up, and overall mortality, in patients with DTC. The analysis did not reveal any statistically significant differences (Table 4, Appendix A). By contrast, Świerniak et al. reported that the TT variant did significantly correlate with conventional risk factors, including lymph node metastasis, extrathyroidal invasion, angioinvasion, and stage IV disease, as well as increased overall mortality, in patients with TC. However, the difference in risk was only significant when comparing the TT variant carriers with carriers of the CC and CT genotypes combined. According to Ensembl (https://www.ensembl.org//), the TT variant is present in 31.8% of Europeans overall (Finnish, 25.3%; British, 33%; Iberian, 36.4%; and Toscani, 38.3%), indicating a high frequency in this populations. Moreover, the TT genotype is more common than the ancestral genotype (CC) [32].

## 5. Conclusions

In conclusion, based on our results, the TT variant of the rs966423 polymorphism in the *DIRC3* gene is not associated with a higher risk of developing DTC, a worse prognosis, or increased overall mortality in patients with DTC in a Polish population. To verify the influence of the TT variant on survival in DTC patients, it may be necessary to perform multi-ethnic studies with larger numbers of patients and controls.

## Figures and Tables

**Table 1 cancers-12-00423-t001:** Characteristics of the study group.

Feature	Total (*n* = 1466)
Gender, *n* (%)	
Female	1269 (86.6%)
Male	197 (13.4%)
Median age, years (Q1–Q3; range)	51.0 (39.0–59.8; 15–85)
Age at diagnosis, ≥ 55 years, *n* (%)	560 (38.2%)
Median tumor size, mm (Q1–Q3; range)	9.0 (5.0–16.0; 0.0–130)
TC main type, *n* (%)	
FTC	34 (2.3%)
Oxyphilic	4 (0.3%)
PDTC	42 (2.9%)
PTC	1386 (94.5%)
Multifocality, *n* (%)	
No	1088 (74.2%)
Yes	378 (25.8%)
Extrathyroidal extension, *n* (%)	
No	1176 (80.2%)
Minor	267 (18.2%)
Gross	23 (1.6%)
Angioinvasion, *n* (%)	
No	1367 (93.2%)
Yes	99 (6.8%)
Tumor stage, *n* (%)	
pT0	2 (0.1%)
pT1a	859 (58.6%)
pT1b	333 (22.7%)
pT2	168 (11.5%)
pT3a	77 (5.3%)
pT3b	11 (0.8%)
pT4a	12 (0.8%)
pT4b	4 (0.3%)
Node stage, *n* (%)	
N0a	770 (52.5%)
N0b	478 (32.6%)
N1a	150 (10.2%)
N1b	68 (4.6%)
Distant metastasis, *n* (%)	
M0	1432 (97.7%)
M1	34 (2.3%)
TNM stage, *n* (%)	
I	1342 (91.5%)
II	98 (6.7%)
III	4 (0.3%)
IVa	3 (0.2%)
IVb	19 (1.3%)
ATA initial risk, *n* (%)	
Low	959 (65.4%)
Intermediate	416 (28.4%)
High	91 (6.2%)

TC, thyroid cancer; FTC, follicular thyroid carcinoma; PDTC, poorly differentiated thyroid carcinoma; PTC, papillary thyroid carcinoma; ATA, American Thyroid Association; TNM, tumor node metastasis.

**Table 2 cancers-12-00423-t002:** Initial response to therapy and response at follow-up in the study group.

Initial Response to Therapy	*n* (%)
Excellent	1116 (83.1%)
Indeterminate	160 (11.9%)
Biochemical incomplete	32 (2.4%)
Structural incomplete	35 (2.6%)
**Final Follow-Up**	***n*** **(%)**
NED	1226 (91.3%)
Indeterminate	69 (5.2%)
Biochemical incomplete	14 (1.1%)
Structural incomplete	33 (2.4%)
**Follow-Up: Recurrence after NED**	***n*** **(%)**
No	1106 (82.3%)
Yes	10 (0.7%)
**Death**	***n*** **(%)**
No	1445 (98.6%)
TC-unrelated	6 (0.4%)
TC-related	15 (1.0%)
**Median Follow-Up Time, Years (Range)**	5.0 (0.0–40.0)

NED, no evidence of disease; TC, thyroid cancer.

**Table 3 cancers-12-00423-t003:** Frequency of rs966423 variants among the study and control groups.

rs966423 Variants	Study Group *n* = 1466, *n* (%)	Controls *n* = 309, *n* (%)	*p*-Value
CC	322 (22.0%)	54 (17.5%)	
CT	730 (49.8%)	155 (50.2%)	
TT	414 (28.2%)	100 (32.4%)	*p* = 0.140
CC	322 (22.0%)	54 (17.5%)	
CT/TT	1144 (78.0%)	255 (82.5%)	*p* = 0.079
TT	414 (28.2%)	100 (32.4%)	
CC/CT	1052 (71.8%)	209 (67.6%)	*p* = 0.147

**Table 4 cancers-12-00423-t004:** Relationships between rs966423 variants and histopathological and clinical factors.

Feature	TT (*n* = 414)	CC/CT (*n* = 1052)	Total (*n* = 1466)	*p*-Value
Gender, *n* (%)				
Female	366 (88.4%)	903 (85.8%)	1269 (86.6%)	
Male	48 (11.6%)	149 (14.2%)	197 (13.4%)	0.1941
Median age at diagnosis, years (Q1–Q3; range)	51.0 (40.0–60.0; 15.0–85.0)	51.0 (39.0–59.0; 15–85)	51.0 (39.0–59.8; 15–85)	0.836
Age at diagnosis, ≥ 55 years, *n* (%)	162 (39.1%)	398 (37.8%)	560 (38.2%)	0.6453
Median tumor size, mm (Q1–Q3; range)	9.0 (6.0–17.8; 0.0–80.0 )	9.0 (5.0–15.0; 0.0–130)	9.0 (5.0–16.0; 0.0–130 )	0.126
TC main type, *n* (%)				
FTC	14 (3.4%)	20 (1.9%)	34 (2.3%)	
Oxyphilic	2 (0.5%)	2 (0.2%)	4 (0.3%)	
PDTC	12 (2.9%)	30 (2.9%)	42 (2.9%)	
PTC	386 (93.2%)	1000 (95.1%)	1386 (94.5%)	0.2411
Multifocality, *n* (%)				
No	315 (76.1%)	773 (73.5%)	1088 (74.2%)	
Yes	99 (23.9%)	279 (26.5%)	378 (25.8%)	0.3042
Extrathyroidal extension, *n* (%)				
No	333 (80.4%)	843 (80.1%)	1176 (80.2%)	
Minor	73 (17.6%)	194 (18.4%)	267 (18.2%)	
Gross	8 (1.9%)	15 (1.4%)	23 (1.6%)	0.7424
Angioinvasion, *n* (%)				
No	381 (92.0%)	986 (93.7%)	1367 (93.2%)	
Yes	33 (8.0%)	66 (6.3%)	99 (6.8%)	0.2437
Tumor stage, *n* (%)				
pT0	1 (0.2%)	1 (0.1%)	2 (0.1%)	
pT1a	227 (54.8%)	632 (60.1%)	859 (58.6%)	
pT1b	102 (24.6%)	231 (22.0%)	333 (22.7%)	
pT2	50 (12.1%)	118 (11.2%)	168 (11.5%)	
pT3a	23 (5.6%)	54 (5.1%)	77 (5.3%)	
pT3b	5 (1.2%)	6 (0.6%)	11 (0.8%)	
pT4a	5 (1.2%)	7 (0.7%)	12 (0.8%)	
pT4b	1 (0.2%)	3 (0.3%)	4 (0.3%)	0.4434
Node stage, *n* (%)				
N0a	208 (50.2%)	562 (53.4%)	770 (52.5%)	
N0b	134 (32.4%)	344 (32.7%)	478 (32.6%)	
N1a	48 (11.6%)	102 (9.7%)	150 (10.2%)	
N1b	24 (5.8%)	44 (4.2%)	68 (4.6%)	0.3479
Distant metastasis, *n* (%)				
M0	402 (97.1%)	1030 (97.9%)	1432 (97.7%)	
M1	12 (2.9%)	22 (2.1%)	34 (2.3%)	0.3552
8th edition of TNM staging, *n* (%)				
I	375 (90.6%)	967 (91.9%)	1342 (91.5%)	
II	31 (7.5%)	67 (6.4%)	98 (6.7%)	
III	2 (0.5%)	2 (0.2%)	4 (0.3%)	
Iva	1 (0.2%)	2 (0.2%)	3 (0.2%)	
Ivb	5 (1.2%)	14 (1.3%)	19 (1.3%)	0.7103
ATA initial risk, *n* (%)				
Low	264 (63.8%)	695 (66.1%)	959 (65.4%)	
Intermediate	118 (28.5%)	298 (28.3%)	416 (28.4%)	
High	32 (7.7%)	59 (5.6%)	91 (6.2%)	0.3018
Response to initial therapy, *n* (%)				
Excellent	319 (83.5%)	797 (82.8%)	1116 (83.1%)	
Indeterminate	45 (11.8%)	115 (12%)	160 (11.9%)	
Biochemical incomplete	9 (2.4%)	23 (2.4%)	32 (2.4%)	
Structural incomplete	8 (2.1%)	27 (2.8%)	35 (2.6%)	0.9528
Final follow-up, *n* (%)				
NED	346 (91.1%)	880 (91.5%)	1226 (91.3%)	
Indeterminate	25 (6.6%)	44 (4.6%)	69 (5.1%)	
Biochemical incomplete	4 (1.1%)	10 (1.0%)	14 (1.1%)	
Structural incomplete	5 (1.2%)	28 (2.9%)	33 (2.5%)	0.280
Follow-up: recurrence after NED, *n* (%)				
No	317 (99.4%)	789 (99%)	1106 (99.1%)	
Yes	2 (0.6%)	8 (1.0%)	10 (0.9%)	0.7852
Death, *n* (%)				
No	408 (98.6%)	1037 (98.6%)	1445 (98.6%)	
TC-unrelated	3 (0.7%)	3 (0.3%)	6 (0.4%)	
TC-related	3 (0.7%)	12 (1.1%)	15 (1.0%)	0.9729
Median follow-up time, years (range)	5.0 (0.0–27.0)	5.0 (0.0–40.0)	5.0 (0.0–40.0)	0.360

TC, thyroid cancer; FTC, follicular thyroid carcinoma; PDTC, poorly differentiated thyroid carcinoma; PTC, papillary thyroid carcinoma; ATA, American Thyroid Association; TNM, tumor node metastasis; NED, no evidence of disease.

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
