# Peer review of "Does the TT Variant of the rs966423 Polymorphism in DIRC3 Affect the Stage and Clinical Course of Papillary Thyroid Cancer?"

_cancers, 2020, doi:10.3390/cancers12020423_

Round 1

Reviewer 1 Report

The manuscript by Hincza and colleagues refers to a study where is evaluate the role of one DIRC3 gene polymorphism that was identified within a long non-coding RNA (lncRNA).

The biology of lncRNA have been gained interest in scientific community, what is a most value of this manuscript.

The results here describe are well documented even being null results.

I do not have major comments but I would like to hear authors about a two minor points.

My main question is why this gene? Apparently, and as was stated by the authors the previous results have been ambiguous, what are the main evidences that support authors' chosen  DIRC3 in thyroid cancer?

Reviewer 2 Report

GWAS identified the DIRC3 gene in thyroid Cancer, but its significant association with thyroid cancer is not clear. Since 2016, only several studies have been published on the DIRC3 gene in thyroid cancer. However, its functional role in thyroid cancer is unknown.

Authors conducted the study to evaluate the impact of the rs966423 polymorphism in the DIRC3 gene in 1,466 DTC patients. However, events such as recurrence and mortality occur in less than 1% in this study, so it is very difficult to find an association between its polymorphism and clinical features. In addition, most patients were low-risk or intermediate-risk patients, making it more difficult to find an association. Therefore, I recommend to conduct sub-group analysis, although the number of high-risk patients is small. If authors have tested other genes, such as BRAF gene or TERT promoter mutation, I recommend that authors also analyze the relationships with them.

Reviewer 3 Report

In the manuscript HiÅ„cza et al. the authors evaluated the impact of the DIRC3 gene polymorphism rs966423 in differentiated thyroid cancer (DTC). The authors also analyzed its possible relationship on mortality, and on some unfavorable histopathological and clinical features. As result of their study including 1466 patients they conclude that there is no significant association between the DIRC3 gene polymorphism rs966423 and any examined histopathological or clinic factor.

Overall the manuscript has the merit of contributing to elucidate the such relationship in the polish population.

I have the following comments:

1) the patients examined mostly presented PTC, therefore the conclusion presented in the manuscript can only be applied to this specific cancer (as the authors correctly specified in the title). however in the conclusion they refer to DTC in general. in my opinion this could generate confusion.

2) the study lasted for 5 years, therefore it is not granted that an association could not be found with a longer followup.
